# Multi-Type Reserve Collaborative Optimization for Gas-Power System Constrained Unit Commitment to Enhance Operational Flexibility

**Jinhao Wang [1], Huaichang Ge [2], Zhaoguang Pan [2,*], Haotian Zhao [2], Bin Wang [2] and Tian Xia [2]**

[1] Electric Power Research Institute of State Grid Shanxi Electirc Power Company, Taiyuan 030001, China; 7wjh@163.com
[2] Department of Electrical Engineering, Tsinghua University, Beijing 100084, China; 13261454982@163.com (H.G.); zhaohaotian@tsinghua.edu.cn (H.Z.); wb1984@tsinghua.edu.cn (B.W.); summersummer@tsinghua.edu.cn (T.X.)
[*] Correspondence: panzg09@163.com

**Abstract:** With the wide application of the gas-power system, gas-power coupling equipment such as gas turbines are gradually becoming widely used, and the problem of insufficient system reserve capacity needs to be solved. In order to improve the operational flexibility of the gas-power system, this paper combines the source side, the load side, and the energy storage side to propose a multi-type backup system, and constructs the source-load-storage multiple reserve capacity system of the gas-power system. Through the participation of gas turbine, steam turbine, interruptible load, and energy storage battery to provide reserve capacity, it can fully cope with the output fluctuation of the load side and source side and realize the coordinated operation of multiple resources to provide reserve capacity. In addition, the coordination of gas turbines and steam turbines can further improve the operation flexibility of the gas-power system. Through the example analysis, it was found that the proposed method reduced the total operating cost of the system by 10.6%.

**Keywords:** gas-power system; reserve capacity; gas turbine; coordinated operation





## 1. Introduction

In recent years, renewable energy has developed rapidly. At present, China has become the country with the largest installed capacity of wind power in the world [1,2]. By the end of 2022, the cumulative installed capacity of wind power in China was 395.6 W. However, the current wind power prediction level is limited, and the randomness and volatility of wind power output pose challenges to the traditional deterministic scheduling method. At the same time, the frequent natural disasters in recent years may cause multiple power equipment failures, which will lead to power system load loss in severe cases. For example, in March 2022, a blackout occurred in Taiwan, resulting in power outages for about 549 million users [3,4]. The uncertainty of wind power output will further aggravate the harm of forced outage of equipment to power systems. Therefore, the power system needs more flexible scheduling methods to cope with the risks caused by uncertainty.

A generator set reserve is an effective means to ensure the scientific operation of the power system [5]. The traditional unit reserve strategy is to ensure that the total reserve of the system is greater than a certain limit. However, due to the uncertainty of wind power output and the forced outage of power equipment, the reserve strategy cannot meet the requirements of economy and reliability of power system operation [6]. Researchers have conducted a lot of research on the reserve optimization strategy of power systems. In Ref. [7], a reserve supply curve construction algorithm based on robust energy and reserve scheduling is proposed. In Ref. [8], a distributed robust formula based on the concept of conditional value at risk (CVaR) is proposed to solve the reserve demand for wind

power. However, the above research only considers how the generator sets provide reserve capacity, which cannot be used in the case of large fluctuations in wind power output.

The gas-power system has been applied more and more in China [9]. By the end of 2022, the number of gas-power systems in China was about 5000. The emergence of gas turbines has increased the power system and natural gas system. A large amount of hot steam will be generated during its operation, which can provide raw materials for steam turbines [10,11]. As a typical device in the gas-power system, the coordinated operation of gas turbines and steam turbines also brings additional operational flexibility to the system. In Ref. [12], in the form of an aggregation model, how to minimize carbon dioxide emissions while providing power and heat loads for gas turbines and steam turbines was studied. Refs. [13,14] establish a configuration-based gas turbine and steam turbine model, which can approximately meet the minimum start–stop constraints of a single turbine and meet other physical constraints of the turbine. In addition, gas turbines and steam turbines together provide spare capacity, which can provide a new method for the flexible operation of gas-power system. Ref. [15] proposes a steam turbine and gas turbine reserve model to solve the uncertainty of renewable energy and realizes the flexible operation of the system. In Ref. [16], the participation of steam turbines and gas turbines in the reserve is considered.

In the gas-power system, in addition to the reserve capacity provided by the source side equipment, the load side and the energy storage side can also provide the reserve capacity for the system [17]. As a kind of power resource to be excavated, interruptible load can be applied to the integrated energy management and microgrid system of the park. The interruptible load on the load side can provide the spinning reserve capacity for the system by reducing the load. The application of energy storage power supply in renewable energy gas-power system provides a feasible solution for the realization of clean and sustainable energy supply [18]. Through the rational design and configuration of the energy storage power system, the reserve capacity of the gas-power system is provided, and the energy management and efficient utilization of resources is guaranteed [19,20].

Therefore, this paper constructs a power generation-load-storage multi-reserve capacity system based on gas power system to solve the problem of system operation fault caused by only the reserve capacity provided by the generator set. The gas power generation system considers gas turbines, steam turbines, interruptible loads, and energy storage batteries to provide reserve capacity. In addition, the cooperative operation of the gas turbine and steam turbine solves the limitation of the traditional gas-power system which only provides flexible resources by gas turbine. The specific contributions of this paper are as follows:

(1) This paper first constructs a source load storage multiple reserve capacity system in the gas-power system, achieving collaborative optimization of multiple types of reserves.

(2) This paper considers the collaborative operation of gas turbines and steam turbines and achieves flexible operation of the system through the combination of different modes.

The structure of the paper is as follows: Section 2 describes the optimal scheduling model of the gas-power system. Section 3 presents the solution strategy of the model. Section 4 describes the case studies. Section 5 is the conclusion.

## 2. Optimal Scheduling Model of Gas-Power System

Figure 1 shows the architecture of the gas-power system. Among them, gas turbine (GT), steam turbine (ST), and power-to-gas (P2G) units are coupling equipment of power system and natural gas system. GTs and ST are power sources. In this paper, the combination of two GTs and one ST is adopted. Because the operation of a steam turbine mainly relies on the waste heat and smoke generated by the gas turbine, the power output of the steam turbine depends on the operating status of the gas turbine. GT can convert natural gas into electric energy and generate the steam required for ST power generation. The coordinated operation of GT and ST can further improve the flexibility of the system to

provide backup. P2G can convert electrical energy into natural gas. In addition, the system also includes energy storage devices and renewable energy.

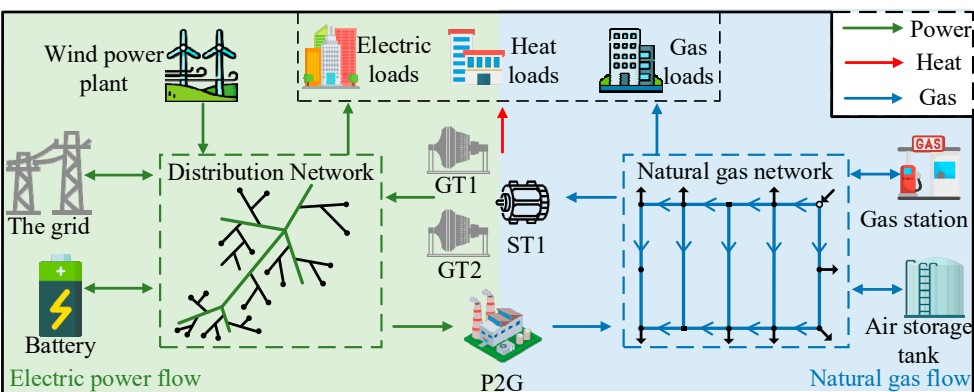

**Figure 1.** The Structure of gas-power system.

Table 1 shows the operating modes of GTs and ST. The operating states of gas turbine and steam turbine form different operating modes. For example, when a GT runs with 0 STs, the system is in mode 2. When a GT runs with 1 ST, the system is in mode 4. The above operation mode enables the gas turbine and steam turbine to cope with the system reserve capacity demand and operation flexibility demand under the condition of renewable energy output fluctuation and load fluctuation.

**Table 1.** Different operation modes of GT and ST.

| Combination | 0 GT + 0 ST | 1 GT + 0 ST | 2 GT + 0 ST | 1 GT + 1 ST | 2 GT + 1 ST |
|:-----------:|:-----------:|:-----------:|:-----------:|:-----------:|:-----------:|
| Mode | 1 | 2 | 3 | 4 | 5 |

*2.1. Multi-Reserve Capacity System of Gas-Power System*

The gas-power system consists of a power generation unit, energy storage device, and interruptible load. Therefore, the gas-power system source-load-storage multiple reserve capacity system can be constructed from the power supply side, the energy storage side, and the load side, as shown in Figure 2. The power supply side mainly relies on GT and ST to provide reserve capacity; the battery in the energy storage device can act as the standby power supply of the system; interruptible load can also provide reserve capacity for the system.

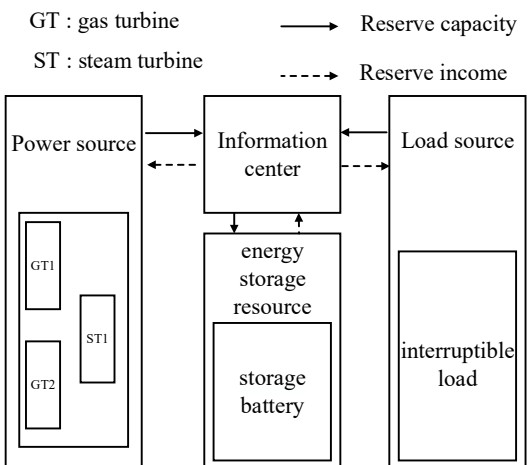

**Figure 2.** Reserve capacity system of gas-power system.

## 2.2. Objective Function

The objective function of the model proposed in this paper is to minimize the operating cost of the gas power generation system, including fuel cost, start–stop cost, and reserve capacity benefit.

$$\min \sum_{t \in T} \left( C_{s,t} + C_{f,t} - C_{r,t}^{GT} - C_{r,t}^{L} - C_{r,t}^{ES} \right) \tag{1}$$

where $C_{s,t}$ is the start–stop cost of gas turbines. $C_{f,t}$ is the cost of natural gas used for the gas-power system. $C_{r,t}^{GT}, C_{r,t}^{L}, C_{r,t}^{ES}$ provide reserve income for gas turbines, interruptible load, and energy storage batteries, respectively.

### 2.2.1. Start–Stop and Operating Costs

$$C_{s,t} = \sum_{i \in \Im^{GT}} \left( c_s^i x_t^i + c_0^i u_t^i \right) \tag{2}$$

where $c_s^i, c_0^i$ are the operating cost and start-up cost of the gas turbines, respectively. $x_t^i, u_t^i$ are the operating binary variables and the starting binary variables of the gas turbines, respectively. $\Im^{GT}$ is the set of gas turbines in the system.

### 2.2.2. Fuel Costs

$$C^f = \alpha^f \sum_{i \in \Im^{GT}} Q_t^i \tag{3}$$

where $\alpha^f$ is the natural gas price. $Q_t^i$ is the amount of natural gas consumed by the gas turbine.

### 2.2.3. Reserve Capacity Revenue

$$C_{r,t}^{GT} = \sum_{i \in \Im^{GT}} \left( c_-^i \cdot R_{t,-}^i + c_+^i \cdot R_{t,+}^i \right) \tag{4}$$

$$C_{r,t}^{ES} = \sum_{i \in \Im^{ES}} \left( c_+^i \cdot R_{t,+}^i + c_-^i \cdot R_{t,-}^i \right) \tag{5}$$

$$C_{r,t}^{L} = \sum_{i \in \Im^{L}} c_+^i \cdot R_{t,+}^i \tag{6}$$

where $c_-^i, c_+^i$ represent the benefits of providing down-spin and up-spin reserves. $R_-^i, R_+^i$ represent the down-spinning reserve and up-spinning reserve provided by the equipment. $\Im^{ES}$ is the set of batteries in the system. $\Im^{IL}$ is the set of interruptible loads in the system.

## 2.3. Operation Constraints of Power System

### 2.3.1. Constraints of Gas Turbines

Gas turbines are a flexible resource in the gas-power system, which closely connects the power system to the natural gas system. Gas turbines have been increasingly used in electrical coupling systems due to their high electrical efficiency and low carbon emissions. Gas turbines' constraints include power output constraints and start–stop constraints. In addition, gas turbines will generate a large amount of steam during operation, which can drive the steam turbine to generate electricity [21].

$$a \left( P_t^{GT} \right)^2 + b P_t^{GT} + c = H_t Q_t^{GT}, t \in T \tag{7}$$

$$x^{GT} \underline{P}^{GT} \leq P_t^{GT} \leq x^{GT} \overline{P}^{GT}, t \in T \tag{8}$$

$$-\Delta \underline{P}^{GT} \leq P_t^{GT} - P_{t-1}^{GT} \leq \Delta \overline{P}^{GT}, t \in T \tag{9}$$

$$u_t^{GT} + v_t^{GT} \leq 1, t \in T \tag{10}$$

$$u_{t+1}^{GT} - v_{t+1}^{GT} = x_{t+1}^{GT} - x_t^{GT}, t \in T \tag{11}$$

$$j\left(P_t^{GT}\right)^2 + kP_t^{GT} + l = W_t^{GT}, t \in T \tag{12}$$

$$0 \leq R_{t,-}^{GT} \leq \min\left\{\left(P_t^{GT} - x_t^{GT} \cdot \underline{P}^{GT}\right), x_t^{GT}\left(\tau \cdot \Delta \underline{P}^{GT}\right)\right\} \tag{13}$$

$$0 \leq R_{t,+}^{GT} \leq \min\left\{\left(x_t^{GT} \cdot \overline{P}^{GT} - P_t^{GT}\right), x_t^{GT}\left(\tau \cdot \Delta \overline{P}^{GT}\right)\right\} \tag{14}$$

where $a, b, c$ are the output coefficients of gas turbine, respectively. $H_t$ is the calorific value of natural gas. $Q_t^{GT}$ is the amount of natural gas consumed by the gas turbine. $\underline{P}^{GT}, \overline{P}^{GT}$ are the minimum/maximum output of gas turbine, respectively. $\Delta \overline{P}^{GT}, \Delta \underline{P}^{GT}$ are the uphill and downhill climbing rates of the gas turbine, respectively. $v_t^{GT}$ is the stopping variable of gas turbine. $j, k, l$ are the steam generation coefficient of gas turbine, respectively. $W_t^{GT}$ is the amount of steam produced by the gas turbine. $R_{t,+}^{GT}, R_{t,-}^{GT}$ are the up-spin reserve and down-spin reserve available for gas turbine, respectively.

### 2.3.2. Constraints of Steam Turbines

The steam turbine (ST) uses the steam generated by the gas turbine to drive the rotor blades to generate electricity. There is also a certain relationship between the steam consumption of steam turbine and the steam production of the gas turbine [22]. The coordinated operation of the gas turbine and steam turbine will further improve the operation flexibility of the gas-power system.

$$P_t^{ST} = f\left(\left(W_t^{ST}\right)^2, W_t^{ST}\right), t \in T \tag{15}$$

$$x^{ST}\underline{P}^{ST} \leq P_t^{ST} \leq x^{ST}\overline{P}^{ST}, t \in T \tag{16}$$

$$-\Delta \underline{P}^{ST} \leq P_t^{ST} - P_{t-1}^{ST} \leq \Delta \overline{P}^{ST}, t \in T \tag{17}$$

$$u_t^{ST} + v_t^{ST} \leq 1, t \in T \tag{18}$$

$$u_{t+1}^{ST} - v_{t+1}^{ST} = x_{t+1}^{ST} - x_t^{ST}, t \in T \tag{19}$$

$$W_t^{ST} + H_t^{HL} \leq W_t^{GT}, t \in T \tag{20}$$

$$0 \leq R_{t,-}^{ST} \leq \min\left\{\left(P_t^{ST} - x_t^{ST} \cdot \underline{P}^{ST}\right), x_t^{ST}\left(\tau \cdot \Delta \underline{P}^{ST}\right)\right\} \tag{21}$$

$$0 \leq R_{t,+}^{ST} \leq \min\left\{\left(x_t^{ST} \cdot \overline{P}^{ST} - P_t^{ST}\right), x_t^{ST}\left(\tau \cdot \Delta \overline{P}^{ST}\right)\right\} \tag{22}$$

where $P_t^{ST}$ is the power output of the steam turbine. (18) and (19) are the relationship between start–stop variables and state variables of steam turbine. (20) indicates that the sum of the steam consumption of the steam turbine and the heat load is less than the steam

production of the gas turbine. $R_{t,+}^{ST}, R_{t,-}^{ST}$ are the up-spin reserve and down-spin reserve available for steam turbine, respectively.

### 2.3.3. Constraints of Batteries

Battery energy storage is a new energy storage technology with high technology maturity at present. It has a fast response speed and the ability to provide fast spinning reserve. The battery plays a role in peak shaving and valley filling in the gas-power system. At the same time, the battery can also provide reserve for the system.

$$V_t = V_{t-1} + \left( \delta^c P_t^c - P_t^{dc}/\delta^{dc} \right) \Delta t \tag{23}$$

$$b^{c/dc} \underline{P}^{c/dc} \leq P_t^{c/dc} \leq b^{c/dc} \overline{P}^{c/dc}, t \in T \tag{24}$$

$$b_t^c + b_t^{dc} \leq 1, t \in T \tag{25}$$

$$\underline{V} \leq V_t \leq \overline{V}, t \in T \tag{26}$$

$$P_t^{dc} + R_{t,+}^e \leq \min\left( \frac{V_t - \underline{V}}{\Delta t} \delta^{dc}, P_0^e \right) \tag{27}$$

$$P_t^c + R_{t,-}^e \leq \min\left( \frac{\overline{V} - V_t}{\Delta t \cdot \delta^c}, P_0^e \right) \tag{28}$$

where $\delta^c, \delta^{dc}$ is the charge/discharge efficiency of the battery. $\underline{P}^{c/dc}, \overline{P}^{c/dc}$ are the battery charge/discharge range. $\overline{V}, \underline{V}$ are the upper/lower limits of battery capacity. $R_{t,+}^e, R_{t,-}^e$ are the up-spin reserve and down-spin reserve available for the battery, respectively. $P_0^e$ is the rated value of battery capacity.

### 2.3.4. Constraints of Interruptible Load

In the gas-power system, the interruptible load can also provide the spinning reserve for the system to realize the flexible operation of the system.

$$0 \leq P_t^{IL} + R_{t,+}^{IL} \leq \overline{P}^{IL}, t \in T \tag{29}$$

$$P_t^{IL} \geq 0, R_{t,+}^{IL} \geq 0, t \in T \tag{30}$$

where $\overline{P}^{IL}$ is the maximum power of the interruptible load. $R_{t,+}^{IL}$ is the upper spinning reserve provided by the interruptible load for the system.

### 2.3.5. Constraints of P2G

P2G is an important part of the gas-power system, which can convert power into natural gas when the electricity price is low, to realize the optimal operation of the system [23].

$$Q_t^g = \delta^g \cdot P_t^g / HHV^g, \forall t \in T \tag{31}$$

$$0 \leq P_t^g \leq \overline{P}^g, \forall t \in T \tag{32}$$

where $\delta^g$ is the power-to-gas efficiency. $HHV^g$ is the high calorific value of natural gas. $\overline{P}^g$ is the maximum power of P2G.

### 2.3.6. Constraints of Power Balance

The power balance constraint ensures that the system runs in a stable state.

$$\sum_{i \in \Im^{GT} \cap \Im^{ST}} P_t^i + \sum_{w \in \Im^{WD}} P_t^w + \sum \left( P_t^{dc} - P_t^c \right) = \sum_{i \in \Im^{p2g}} P_t^i + P_t^{EL} - P_t^{IL}, \forall t \in T. \tag{33}$$

where $\Im^{WD}$ is the set of wind power plants. $\Im^{p2g}$ is the set of P2Gs. $P_t^{EL}$ is the electric load power.

### 2.4. Operation Constraints of Natural Gas System

The natural gas network model adopts the steady-state Weymouth equation.

$$Q_t^q + \sum_{pq \in X(q)} Q_t^{pq} = \sum_{qk \in Y(q)} Q_t^{qk}, t \in T \tag{34}$$

$$Q_t^q = Q_t^{q,well} + \sum_{g \in \Im_q^{p2g}} Q_t^{q,g} - Q_t^{q,l} - \sum_{i \in \Im_q^{GT}} Q_t^i, t \in T \tag{35}$$

$$Q_t^{pq} = C^{pq} \sqrt{\left| (\varpi^p)^2 - (\varpi^q)^2 \right|}, t \in T \tag{36}$$

where $p, q$ represent the start node and the end node of the natural gas network, respectively. $X(q), Y(q)$ represent the set of starting nodes and ending nodes, respectively. $\varpi^p, \varpi^q$ represent the pressure of the starting node and the ending node, respectively. $Q_t^{q,well}$ is the gas source power output.

## 3. Solution Methodology

There are quadratic terms of power generation in Equations (7) and (12) in this paper, which can be linearized by cone constraints and introducing auxiliary variables [21]. The Formula (38) is the second-order cone relaxation form of Formula (37).

$$\pi_t = \left( P_t^{GT} \right)^2, t \in T \tag{37}$$

$$\left\| \begin{array}{c} \sqrt{2} P_t^{GT} \\ \pi_t \\ 1 \end{array} \right\|_2 \leq \pi_t + 1, t \in T \tag{38}$$

where $\pi_t$ is an auxiliary variable introduced by convex quadratic term linearization. The Formula (38) is a linear second-order cone relaxation constraint.

The natural gas system is mainly composed of the natural gas source, natural gas pipeline, and natural gas load. In order to reduce the difficulty of solving the natural gas system model and reduce the complexity of the coupling between the power system and the natural gas system, the nonlinear natural gas pipeline flow equation is linearized by the piecewise linearization method, and then the existing natural gas subsystem model is transformed into a mixed integer linear programming model. The schematic diagram of piecewise linearization is shown in Figure 3.

It should be noted that due to the natural gas in the transmission process by their own materials and external factors, a pressure drop will be produced. In order to maintain the node pressure at a normal level and reduce the probability of gas transmission blockage in the natural gas pipeline during the peak gas load, it is necessary to install compressors along the natural gas pipeline. Because the compressor consumes less energy, in order to simplify the calculation, only the node pressure relationship at both ends of the compressor is retained, and the energy consumed by the compressor is not considered.

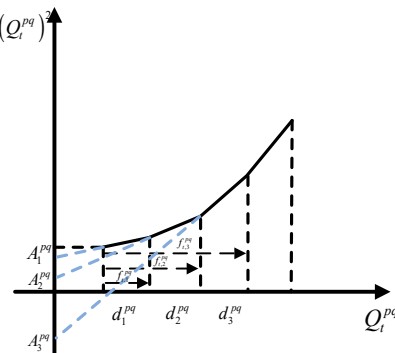

**Figure 3.** Piecewise linearization diagram of the natural gas network.

$$\left(Q_t^{pq}\right)^2 = (C^{pq})^2 \cdot (\varpi s^p - \varpi s^q), t \in T \tag{39}$$

$$\left(Q_t^{pq}\right)^2 = \sum_{l}^{NL} \left(f_{t,l}^{pq} \cdot \beta_l^{pq} + A_l^{pq} \cdot \delta_l^{pq}\right), t \in T \tag{40}$$

$$0 \le f_{t,l}^{pq} \le \delta_l^{pq} d_l^{pq}, l = 1, t \in T \tag{41}$$

$$\delta_l^{pq} d_{l-1}^{pq} \le f_{t,l}^{pq} \le \delta_l^{pq} d_l^{pq}, l \ge 2, t \in T \tag{42}$$

$$\sum_{l=1}^{NL} \delta_l^{pq} = 1 \tag{43}$$

$$\sum_{l=1}^{NL} f_{t,l}^{pq} = V_t^{pq} \tag{44}$$

$$d_l^{pq} = \frac{Q_{\max}^{pq} l}{m} \tag{45}$$

The linearization process of the natural gas network is shown in Formulas (39)–(45). By dividing the abscissa $Q_t^{pq}$ into m parts, *m* primary curves are constructed. Formula (39) is the square of the Weymouth equation. Formula (40) is piecewise linearization. The Formulas (41)–(45) are used to judge that the natural gas system works in a certain segmentation interval.

## 4. Case Studies

All tests were resolved using CPLEX interfaced through MATLAB 2021a [23], and the experimental running computer environment was 11th Gen Intel (R) Core (TM) i7-1165G7 @ 2.80 GHz dual-core processor, 16 GB RAM, and Windows 10 system.

### 4.1. Test System Description

The example scenario constructed in this paper includes two gas turbines, a steam turbine, a P2G unit, an energy storage battery, and a wind power plant. The purpose of this paper is to study the operation of a gas-power system with multi-type reserve participation. The load, wind power forecasting, and reserve demand in this paper are shown in Figure 4.

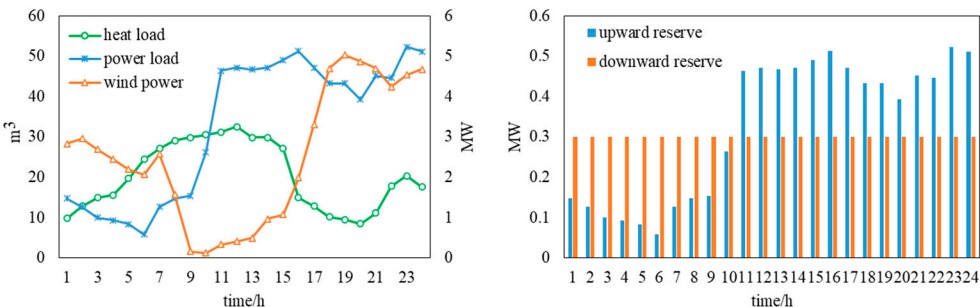

**Figure 4.** The load, wind power forecasting, and reserve demand.

### 4.2. Case Setting and Result Analysis

The following three cases are investigated to compare and analyze the results:

Case 1: Gas turbines and steam turbines are involved in providing reserve capacity.

Case 2: Gas turbines, steam turbines, and interruptible loads are involved in providing reserve capacity.

Case 3: Gas turbines, steam turbines, interruptible loads, and energy storage batteries are involved in providing spare capacity.

The scheduling results of the three cases are shown in Table 2.

**Table 2.** Costs and revenues in case 1 to case 3.

|        | $C_f$    | $C_s$ | $C_r$  | Total Cost |
|--------|----------|-------|--------|------------|
| Case 1 | 22,718.3 | 300   | 618.5  | 22,399.8   |
| Case 2 | 22,805.4 | 300   | 860.5  | 21,644.9   |
| Case 3 | 21,107.3 | 250   | 1345.7 | 20,011.6   |

It can be seen from the scheduling results that the total cost of the system is reduced when the gas turbine, steam turbine, interruptible load, and energy storage battery participate in the provision of reserve capacity at the same time.

#### 4.2.1. Analysis of Device Power Output

The device power output is shown in Figures 5–7.

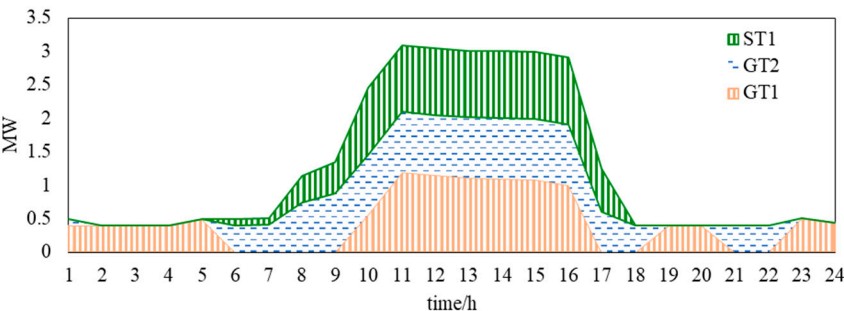

**Figure 5.** Unit power output in case 1.

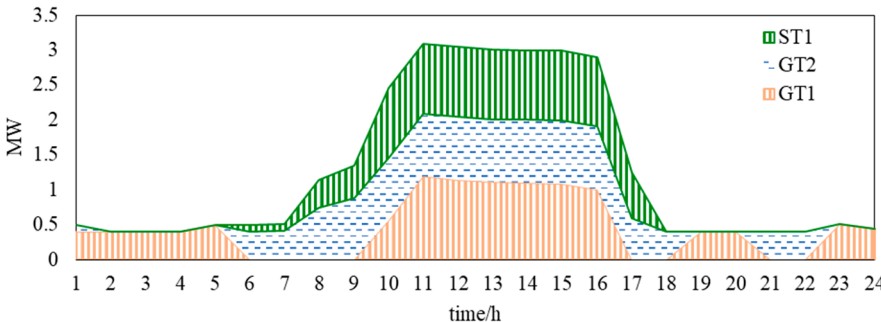

**Figure 6.** Unit power output in case 2.

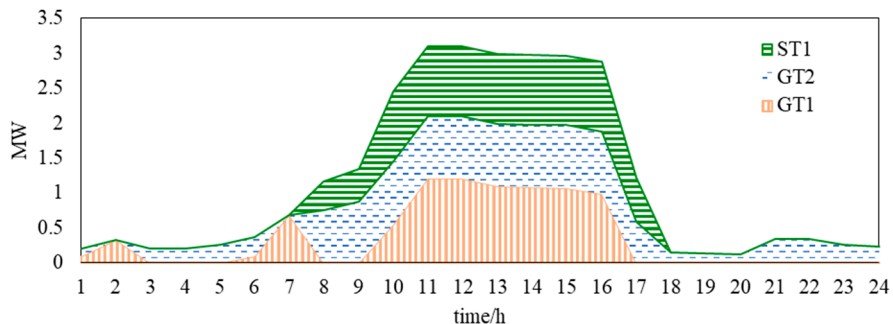

**Figure 7.** Unit power output in case 3.

From the Figures 5–7, it can be seen that compared with cases 1 and 2, case 3 changes the operation mode of the unit during the period of low load and high wind power. During the period of 3–5, because the power generation cost of GT1 is less than that of GT2, case 1 chooses GT1 to generate power to take into account the reserve demand and heat load demand, and the power generation is higher. In case 3, due to the participation of energy storage batteries in providing backup, the amount of steam generated by GT2 is greater than that of GT1 under the same power generation. Therefore, the small amount of electricity generated by GT2 can meet the heat load demand of the period and take into account the backup demand. Although the power generation cost of GT2 is high, the power generation is small, and the cost is still less than that of case 1. The reasons for the change of unit operation mode in the 19–20 and 23–24 periods are the same as above. In the 6–7 period, the heat load level is high. To meet the load demand and reserve demand, GT2 + ST1 are selected to operate together, and the output of GT2 is at a high level. In case 3, the unit does not need to consider the reserve demand and only needs to select the best operation mode that can meet the electrical load and thermal load at the same time.

In summary, the participation of battery energy storage in reserve makes the total operating hours of the unit less than cases 1 and 2, and the number of start-stops is reduced. At the same time, the total power generation of the unit is reduced, and the fuel cost is reduced, indicating that the participation of battery energy storage in reserve will increase the operational flexibility of the system, making the unit generate electricity in a more economical operation mode.

### 4.2.2. Analysis of Device Operation Modes

Table 3 shows the operation mode results of GTs and ST in different periods.

**Table 3.** The device operation mode of case 1 to case 3.

| Hour | 1 | 2 | 3 | 4 | 5 | 6 | 7 | 8 | 9 | 10 | 11 | 12 |
|---|---|---|---|---|---|---|---|---|---|---|---|---|
| Mode (Case 1) | 4 | 2 | 2 | 2 | 2 | 6 | 6 | 6 | 6 | 7 | 7 | 7 |
| Mode (Case 2) | 4 | 2 | 2 | 2 | 2 | 6 | 6 | 6 | 6 | 7 | 7 | 7 |
| Mode (Case 3) | 4 | 2 | 3 | 3 | 3 | 4 | 2 | 6 | 6 | 7 | 7 | 7 |
| **Hour** | **13** | **14** | **15** | **16** | **17** | **18** | **19** | **20** | **21** | **22** | **23** | **24** |
| Mode (Case 1) | 7 | 7 | 7 | 7 | 6 | 3 | 2 | 2 | 3 | 3 | 2 | 2 |
| Mode (Case 2) | 7 | 7 | 7 | 7 | 6 | 3 | 2 | 2 | 3 | 3 | 2 | 2 |
| Mode (Case 3) | 7 | 7 | 7 | 7 | 6 | 3 | 3 | 3 | 3 | 3 | 3 | 3 |

It can be seen from the scheduling results that compared with case 1, case 3 changes the operation mode of the unit during low load and high wind power periods.

(1) In the periods of 3–5, because the cost of GT1 power generation is less than that of GT2, case 1 chooses GT1 power generation to take into account the reserve demand and heat load demand, and the power generation is higher. In case 3, due to the participation of energy storage batteries, the amount of steam generated by GT2 is greater than that of GT1 under the same power generation. Therefore, a small amount of electricity generated by GT2 can meet the heat load demand of this period and take into account the reserve demand. Although the power generation cost of GT2 is high, the power generation is small, and the cost is still less than that of case 1. The reasons for the change of unit operation mode in the 19–20 and 23–24 periods are the same as above.

(2) In the 6–7 periods, the heat load level is high. To meet the load demand and reserve demand, case 1 chooses GT2 + ST1 to operate together, and the output of GT2 is at a high level. In the case of case 3, the unit does not need to consider the reserve demand and only needs to select the best operation mode that can meet the electrical load and thermal load at the same time.

In summary, the participation of battery energy storage in reserve makes the total operation hours of the unit less than that of cases 1 and 2, and the number of start-ups and stops is reduced. At the same time, the total power generation of the unit is reduced, and the fuel cost is reduced. It is shown that the participation of battery energy storage in reserve will increase the operation flexibility of GTs and ST, making the generator units operate more economically.

### 4.2.3. Analysis of Reserve Optimization

The battery capacity changes of the three schemes are shown in Figure 7. The reserve capacity provided by the system is shown in Figure 8.

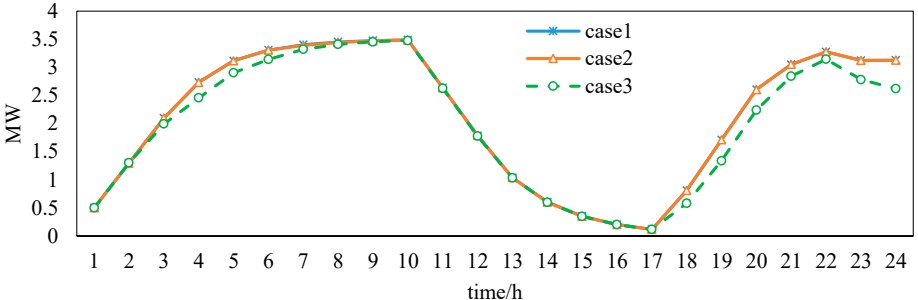

**Figure 8.** Battery capacity in case 1, 2, and 3.

From the Figures 8 and 9, it can be seen that compared with case 1, case 2 can provide an up-spinning reserve for the system through interruptible load during low load periods, which can increase the output of the unit, make the battery store more electricity during low load period, and reduce the load loss during peak load period.

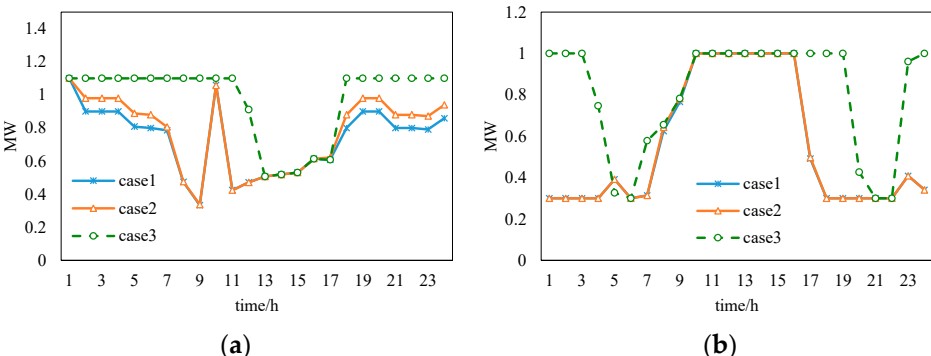

**Figure 9.** Reserve capacity provided by the system: (**a**) upward spinning reserve, (**b**) downward spinning reserve.

In cases 1 and 2, battery energy storage can only participate in peak shaving operation through low storage and high incidence, and only the generator set increases output charging when the load is low. In case 3, after the energy storage battery participates in the reserve, it will reduce the storage capacity in the low-load 1–6 period to provide the down-spin reserve, so that the unit's output at the 1–6 moment will be reduced. It will also provide the down-spin reserve to reduce the unit's output during the period of high wind power after the 18 period. Therefore, the wind curtailment rate is reduced, and more spinning reserves will be set aside, which increases the reserve income.

## 5. Conclusions

In this paper, a multi-type reserve capacity system of source-load-storage was constructed based on the gas-power system. Through the example results, when the gas turbine, steam turbine, interruptible load, and energy storage battery participate in the reserve, the operating cost of the system is reduced by 10.6%. In addition, the cooperative operation mode of the gas turbine and steam turbine makes the fuel cost of the system decrease by 7.1%, the start and stop times of the generator set decrease by one time, the overall operation time decrease, and the operation flexibility of the gas-power system increase.

In the future, we will further tap the reserve resources of the gas-power system and improve the operational flexibility of the power system. In addition, we also plan to study the operation of the gas-power system under multiple uncertainties to improve the system's ability to resist risks.

**Author Contributions:** Conceptualization, B.W. and T.X.; Methodology, Z.P., B.W. and T.X.; Data curation, H.G.; Writing—original draft, Z.P. and H.Z.; Writing—review & editing, H.G., Z.P. and H.Z.; Supervision, J.W. and H.G.; Project administration, J.W.; Funding acquisition, J.W. All authors have read and agreed to the published version of the manuscript.

**Funding:** This research is funded by the Science and Technology Project of Shanxi Electric Power Company No. 52053022000K.

**Conflicts of Interest:** The authors declare no conflict of interest.

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
