# Peer review of "Multi-Type Reserve Collaborative Optimization for Gas-Power System Constrained Unit Commitment to Enhance Operational Flexibility"

_electronics, doi:10.3390/electronics12194029_

Round 1

Reviewer 1 Report

The authors reported "Coordinated Optimization of Multi-type Reserve in gas-power system". The manuscript comprises series of investigation with relevant descriptions. It can be considered after considering the following major points:

1. The Abstract should reflect the main finding with some required values.

2. What is new in this report? Please provide the gaps that the present report will expected to fill in introduction part.

3. Section 3, the methodology is not clear.

4. The conclusion is not acceptable in the current form. Please thoroughly revise with sufficient information.

5. Please check the whole manuscript, including language, spacing, etc.

I think the language is fine but needs some revision.

Reviewer 2 Report

This paper is of good academic quality and has a good potential for publication in this journal. Only minor revision is suggested.

(1) in pp. 8, line 257, section 4.2.1 should be section 4.2.2, because there is already a sectin 4.2.1 in pp. 7

(2) English proof-reading service is suggested.

The overall quality of English language is acceptable, however, minor editing of English language is suggested via language proof-reading service. 

Reviewer 3 Report

The abstract is very brief.

How can the issue of inadequate system reserve capacity in gas-power systems be effectively addressed while enhancing operational flexibility through coordinated resource utilisation? This paper proposes a multi-type backup system incorporating gas and steam turbines, interruptible loads, and energy storage batteries.

The introduction highlights a movement towards taking gas-powered systems into consideration and notes the conventional limits of generator set reserve techniques. How does the proposed method create a multi-reserve capacity system for the gas-power system and how does this support both reliability and efficiency in power system operation? How does the proposed method leverage the capabilities of gas turbines, steam turbines, interruptible loads, and energy storage batteries?

The introduction does not describe the problem statement in detail per the proposed work.

The mathematical model used or given in the article is already available in the literature, but the overall contribution is not clear in the proposed hypothesis?

The results section is weak and does not consider the case studies or multiple case scenarios for the proposed study.

Findings must support the conclusion.

Improve language.

Reviewer 4 Report

The review of the article " Coordinated Optimization of Multi-type Reserve in gas-power  system " is completed.

1.   The abstract not related to main body of paper  

Main body of paper discuss the gas turbine and wind energy (figure 1)- (steam turbines not mention in introduction and model of the system).

2.      the abstract is not clear.

3.      The English must be improved. I recommend a spell-check, and the consultation of a native speaker.

4.      In some figures (figs. 7), all presented curves were displayed by line. It is better to use different types for the curves in one figure (line, dash-line, dash-dash, ……..).

5.      All references from one source (IEEE Trans. Smart Grid), References sources must be diversified

6.      figure 1 not included steam turbine.

7.      In solution methodology part, what mean by this sentence (There are quadratic terms of power generation in (9) and (12) in this paper,).

8.      Present the methods used in figure 1.

9.      I suggest adding the methodology in section 2 (after introduction).

10.   in part 3 (solution methodology) , please remove this sentence (For details, please refer to Ref. 20).

11.   in line 194 and 195

(37)-(43) shows the linearization process of natural gas network.

you should never begin a sentence with a numeral. Instead, you should try to reword the sentence.

12.   In the case study part (part 4), please focus and explain more about the test related to this study.

13.   In line 282, please correct (sys-tem).

14.   The conclusion paragraph should restate your paper, summarize the key supporting ideas you discussed in the paper. in addition to future work.

1.      The English must be improved. I recommend a spell-check, and the consultation of a native speaker.

Round 2

Reviewer 1 Report

The authors addressed all my previous comments. It can be accepted for publication.

Reviewer 3 Report

The revision is not satisfactory, and it is suggested to revise the article very exclusively 

The author need to check for typos.

Reviewer 4 Report

There are no additional comments

Round 3

Reviewer 3 Report

OK

Check-in final version